# 9p21 Locus Polymorphism Is A Strong Predictor of Metabolic Syndrome and Cardiometabolic Risk Phenotypes Regardless of Coronary Heart Disease

**DOI:** 10.3390/genes13122226

**Published:** 2022-11-27

**Authors:** Muhammad Mobeen Zafar, Muhammad Saqlain, Asad Mehmood Raja, Pakeeza Arzoo Shaiq, Muhammad Javaid Asad, Muhammad Kausar Nawaz Shah, Farah Fatima, Hadi Valadi, Muhammad Nawaz, Ghazala Kaukab Raja

**Affiliations:** 1University Institute of Biochemistry & Biotechnology, PMAS-Arid Agriculture University Rawalpindi, Rawalpindi 46300, Pakistan; 2Department of Biochemistry and Molecular Biology, University of Sialkot, Sialkot 51040, Pakistan; 3Department of Plant Breeding and Genetics, PMAS-Arid Agriculture University Rawalpindi, Rawalpindi 46300, Pakistan; 4Department of Pathology, Ribeirao Preto Medical School, University of Sao Paulo, Ribeirao Preto 3900, Brazil; 5Department of Rheumatology and Inflammation Research, Institute of Medicine, Sahlgrenska Academy, University of Gothenburg, 41346 Gothenburg, Sweden

**Keywords:** 9p21 locus, cardiometabolic risk phenotype, coronary heart disease, CHD, metabolic syndrome, obesity

## Abstract

The world population is genetically predisposed to metabolic syndrome (MetS) and its components, also known as cardiometabolic risk phenotypes, which can cause severe health complications including coronary heart disease (CHD). Genetic variants in the 9p21 locus have been associated with CHD in a number of populations including Pakistan. However, the role of the 9p21 locus in MetS and cardiometabolic risk phenotypes (such as obesity, hypertension, hyperglycemia, and dyslipidemia) in populations with CHD or no established CHD has not been explored. Therefore, the present study was designed to explore the association of the minor/risk allele (C) of 9p21 locus SNP rs1333049 with MetS or its risk phenotypes regardless of an established CHD, in Pakistani subjects. Genotyping of rs1333049 (G/C) was performed on subjects under a case-control study design; healthy controls and cases, MetS with CHD (MetS-CHD^+^) and MetS with no CHD (MetS-CHD^−^), respectively. Genotype and allele frequencies were calculated in all study groups. Anthropometric and clinical variables (Means ± SD) were compared among study groups (i.e., controls, MetS + CHD and MetS-CHD) and minor/risk C allele carriers (GC + CC) vs. non-carriers (Normal GG genotype). Associations of the risk allele of rs1333049 SNP with disease and individual metabolic risk components were explored using adjusted multivariate logistic regression models (OR at 95% CI) with a threshold *p*-value of ≤0.05. Our results have shown that the minor allele frequency (MAF) was significantly high in the MAF cases (combined = 0.63, MetS-CHD^+^ = 0.57 and MetS-CHD^−^ = 0.57) compared with controls (MAF = 0.39). The rs1333049 SNP significantly increased the risk of MetS, irrespective of CHD (MetS-CHD^+^ OR = 2.36, *p* < 0.05 and MetS-CHD^−^ OR = 4.04, *p* < 0.05), and cardiometabolic risk phenotypes; general obesity, central obesity, hypertension, and dyslipidemia (OR = 1.56–3.25, *p* < 0.05) except hyperglycemia, which lacked any significant association (OR = 0.19, *p* = 0.29) in the present study group. The 9p21 genetic locus/rs1333049 SNP is strongly associated with, and can be a genetic predictor of, MetS and cardiometabolic risks, irrespective of cardiovascular diseases in the Pakistani population.

## 1. Introduction

The cardiometabolic risk phenotypes/dysregulations such as obesity, hypertension, dyslipidemia, and hyperglycemia (collectively known as metabolic syndrome (MetS)), are the highest non-communicable health risks faced by the world population [1]. Depending upon worldwide population variabilities like sex, age, race/ethnicity, and lifestyle-related factors, MetS prevalence ranges from 10 to 84% [2], while in the South Asian population it ranges from 11.9 to 49%, with the highest rate of 54.9% in Pakistan [3,4]. Cardiometabolic dysregulations and MetS lead to a number of health-threatening issues including coronary heart disease (CHD) [5,6,7,8]. MetS is predicted to increase the risk of CHD episodes by two-fold and mortality by one-and-a-half-fold in the next 5 to 10 years [9,10,11], regardless of gender and age [12]. However, South Asian ethnicities (Bangladesh, Bhutan, India, Maldives, Pakistan, Nepal, and Sri Lanka) are considered to be highly susceptible to cardiometabolic disorders and CHD [12,13,14,15]. In Pakistan, CHD shares the highest incidence rate (21%) compared with other non-communicable diseases with an annual increase of 0.85% [4,16,17].

CHD is a major health complication and impediment to achieving global health goals with a 28% rate of total disease-related deaths, an annual rate of 0.36% [17]. It is a multifactorial and multi-genic disorder involving abnormalities in both the myocardial and vascular processes aggravated by ischemia, inflammation, and apoptosis [18]. Both GWAS and candidate gene studies over the years have reported a number of genetic loci contributing to the development of CHD in different populations [19,20]. Among many genetic loci, the 9p21 locus was the first genetic risk factor having a high effect size and the most replicated one as well [20,21,22,23,24]. 9p21 SNPs have been analyzed to forecast ischemic events e.g., MI amongst CAD patients [19,20].

Among various SNPs on CHR 9p21, rs1333049 (CDKN2A/B gene) has been reported to be linked to the incidence and development of coronary artery disease (CAD) and myocardial infarction (MI) [22,25,26,27,28,29,30,31]. However, it still needs to be explored whether the rs1333049 variant of the 9p21 locus has a direct role in the development of heart diseases or its effect is triggered by existing cardiometabolic dysregulations, including the well-established MetS [20]. With a focus on the South Asian ethnicity with a high prevalence of MetS/components and CHD, the present study was designed to explore the association of the minor C allele of the rs1333049 variant in Pakistani individuals having MetS with or without CHD.

## 2. Materials and Methods

### 2.1. Characteristics of Study Subjects

The study consisted of 400 subjects, 109 healthy controls and 291 cases (208 MetS with CHD (Mets-CHD^+^) and 83 MetS without CHD (Mets-CHD^−^). For MetS, the International Diabetes Federation (IDF) criteria were used [32]. The CHD status of study subjects was confirmed by a cardiologist with at least one cardiac episode during the last year and with/without common cardiometabolic risk phenotypes; general/central obesity, hypertension, hyperglycemia, and dyslipidemia. The CHD patients were sampled from cardiology departments in local hospitals of Rawalpindi, Islamabad and Lahore, Pakistan. Prior approval from the ethics committee for the use of human subjects, PMAS-AAUR, was obtained (protocol number: PMAS-AAUR/IEC-18/2018 and dated 27 September 2018). The study subjects agreed to the sampling by signing the written informed consent. A questionnaire was used for the collection of anthropometric data, clinical and genetic data. The sample size was calculated using the online OpenEpi calculator (http://www.openepi.com) accessed on 12 March 2020.

### 2.2. Anthropometric and Clinical Data Collection

The distribution of individuals in three study groups was as follows: healthy controls 33% (with no history of MetS or CHD), MetS-CHD^+^ 46% and MetS-CHD^−^ 21%. Anthropometric parameters included age, weight, height, waist circumference (WC), hip circumference (HC), and blood pressure (systolic SBP mmHg and diastolic DBP mmHg). The body adiposity indices; body mass index (BMI Kg/m^2^), WC, waist-to-hip ratio (WHR), visceral adiposity index (VAI), and body adiposity index (BAI) were calculated using standard formulas [33].

The clinical test profiles; lipid profile (triglycerides (TG), total cholesterol (TC), low-density lipoprotein (LDL), high-density lipoprotein (HDL), and fasting blood glucose (FBG) were determined using standard laboratory protocols. Cardiometabolic risk phenotypes were calculated using the above data sets; general obesity or GO (BMI ≥ 30 Kg/m^2^), central obesity or CO (WHR > 0.90 for male and 0.85 for females), hypertension (SBP > 139 mmHg and DBP > 89 mmHg) [34], hyperglycemia (FBG > 100 mg/dL) [35], dyslipidemia (LDL > 100 mg/dL, HDL < 40 mg/dL and TG > 150 mg/dL) [36].

### 2.3. DNA Isolation and Genotyping

For genetic analysis, 3 mL of venous blood was collected in EDTA-coated tubes. DNA was isolated using the organic solvent extraction method [37]. The DNA was quantified using NanoDrop1000 and its purity was examined for genetic analysis. For genotyping, allele-specific primers were used for the amplification of C/G alleles of rs1333049 SNP and its flanking regions. For the amplification of C allele, an inner reverse allele-specific (5′-CTTACCTCTGCGAGTGGCTGCTTATG-3′) and outer forward (5′-CGAAGTAGAGCTGCAAAGATATTTGGAA-3′) primers were used, while for G allele an inner forward allele-specific (5′-CCTCATACTAACCATATGATCAACAGATG-3′) and outer reverse (5′-GGGCTCATAATTGCTGAATAAAACAGAA-3′) primers were used. The thermal profile used for PCR was: initial denaturation at 95 °C for 5 min, 34 cycles of denaturation at 95 °C for 45 s, primer annealing at 60 °C for the G allele, and 54.7 °C for the C allele for 60 s, and extension at 72 °C for 45 s. Final extension was executed at 72 °C for 5 min. The PCR amplified products of 263 bp corresponded to the homozygous CC genotype and 214 bp to the homozygous GG genotype, respectively, on 2% agarose gel. The CG heterozygous genotype resulted in two DNA bands of 263 bp and 214 bp.

### 2.4. Statistical Analysis

Genotype and allele frequencies were calculated in all study groups and evaluated for the Hardy Weinberg Equilibrium (HWE) using the chi-square test. The minor C allele frequency was compared between controls and cases using the chi-square test. The association of the minor allele C with age, gender, and disease status; MetS-CHD^+^ or MetS-CHD^−^; and individual cardiometabolic risk phenotypes (GO, CO, hypertension, dyslipidemia, hyperglycemia) was calculated using the dominant genetic model (CC + GC vs. GG). The odds ratios (OR) and 95% confidence intervals (95% CI) were estimated using logistic regression after adjustments for age, sex, BMI, WC, WHR, hypertension, dyslipidemia, and hyperglycemia. A significance level of *p* ≤ 0.05 was used in all statistical analyses. To find the levels of all continuously measured study variables in C allele carriers, the standard deviation (SD) change of each variable and adiposity index from their population means were computed for C (CC + GC) and G (GG) allele carriers. Descriptive statistics (Mean ± SD) were used to summarize the collected data using one-way analysis of variance (ANOVA). Data analysis was carried out using SPSS version 22.0 (SPSS Inc., Chicago, IL, USA).

## 3. Results

### 3.1. Anthropometric Parameters

All study variables were compared (Mean ± SD) among healthy controls and cases (combined). The anthropometric parameters (age, weight, HC, SBP and DBP), adiposity indices (BMI, WC, WHR, VAI, BAI), and clinical test values (FBG, TG, TC, LDL) in all the case groups were in risk ranges and significantly higher (*p* < 0.05) compared with the healthy controls. Whereas HDL levels were significantly lower (*p* < 0.05) in the case groups as compared with controls (Table 1). The frequency distribution (%) of individual cardiometabolic risk phenotypes (dyslipidemia, hypertension, hyperglycemia, CO, GO) was also calculated in all cases. The highest frequency was dyslipidemia (76.7%) followed by GO (56.2%), hypertension (54.8 (53.4%), and CO (44.5%) (Figure 1).

### 3.2. Minor Allele Frequency (MAF) Distribution

The minor allele frequency (MAF) distribution of rs1333049 between the case and control groups was compared to explore the genetic susceptibility of the variant toward disease in a Pakistani population. The frequency of the minor and risk C allele of the rs1333049 variant was higher in individuals from the case groups (combined MAF = 0.63, MetS-CHD^−^ and MetS-CHD^+^ MAF = 0.57, respectively) as compared with the control group (MAF = 0.39, *p*-value < 0.0001). Compared with the controls, the significantly high frequency of minor C allele in the cases clearly indicated it was a risk allele (Figure 2).

### 3.3. Association Analysis

The association analysis of the rs1333049 genetic variant with disease groups MetS-CHD^−^ and MetS-CHD^+^ and individual cardiometabolic risk phenotypes; dyslipidemia, hypertension, hyperglycemia, CO, GO are presented in Table 2. Per the analysis results, individuals carrying minor/risk C allele (GC + CC) had a 4 times higher risk of MetS-CHD^−^ (OR = 4.04, 95% CI = 1.62–10.07) and 2.7 times higher risk of MetS-CHD^+^ (OR = 2.24–2.70, 95% CI = 1.16–6.28) compared with non-carriers (GG). The C allele is also significantly associated with individual cardiometabolic risk phenotypes. The C allele carriers had almost a 3-fold risk of dyslipidemia (OR = 3.25, 95% CI = 2.28–6.07), 2.4-fold GO (OR = 2.39, 95% CI = 1.15–5.00), 2-fold hypertension (OR = 2.24, 95% CI = 1.11–4.53), and 1.6-fold CO (OR = 1.56, 95% CI = 1.26–3.68). The rs1333049 variant also showed an association with gender; the C allele carrier male individuals were at 1.8 times higher risk of disease/cardiometabolic risk phenotypes as compared with females (OR = 1.79, 95% CI = 1.13–3.44). Whereas the hyperglycemia lacked any association with rs1333049 (OR = 0.19, 95% CI = 0.01–4.64).

All study variables (Mean ± SD) were also compared among minor/risk C allele carriers (GC + CC) versus non-carriers (GG) of the rs1333049 variant. The C allele carriers had significantly higher (*p* ≤ 0.05) risk ranges of anthropometric (weight, SBP and DBP) and clinical variables (FBG, TG, TC, HDL, LDL) except HDL, which were significantly lower (*p* ≤ 0.05) in carriers compared to non-carriers (Figure 3). The value (higher or lower) of continuous covariates in risk allele carriers showed that C allele carriers were 2.3 years older and 0.1 Kg heavier in body weight as compared with the G allele carriers. The SBP was 8.4 mm/Hg lower while DBP was 2.2 mm/Hg higher in risk allele carriers.

The FBG, TG, and LDL levels were raised by 2.1 mg/dL, 2.3 mg/dL, and 1.0 mg/dL, respectively; while the TC and HDL were 2.6 mg/dL and 1.7 mg/dL lower, respectively, in risk allele carriers as compared with non-carriers (Figure 4). The combined datasets of Figure 3 and Figure 4 have been included as a Appendix A. Additionally, the adiposity indices BMI, WC, WHR, VAI, and BAI were all raised in the C allele carriers as compared with the G allele carriers (Figure 5). Per SD change, BMI was 0.4 Kg/m^2^, WC 3.9 cm, and VAI 0.1 and BAI 1.4 raised while WHR was 0.009 lower in C allele carriers (Figure 6). The combined datasets of Figure 5 and Figure 6 have been included as a Appendix A.

## 4. Discussion

In the present study, we found that the minor/risk allele (C) of the 9p21 locus SNP rs1333049 showed a strong association with MetS as a whole, as well as with its individual cardiometabolic risk components, irrespective of established CHD in subjects from Pakistan. The minor allele frequency distribution of the rs1333049 SNP was compared between the controls and case groups as MetS^+^ CHD and MetS^−^ CHD. The C allele frequency was significantly higher in all disease groups as compared with controls.

The prevalence of MetS and cardiometabolic risk phenotypes abdominal obesity, dyslipidemia, hypertension, and hyperglycemia are on the rise in adult populations of most countries around the globe [1]; however, children and adolescents are also becoming its victims [20,33,38,39,40]. The major risk factors leading to increasing rates of disease are sedentary and stressful lifestyles coupled with excessive consumption of caloric-rich foods and obesity [5,11,40,41]. MetS and its components have been associated with CAD [4,5,6]. All case subjects of the present study had MetS regardless of CHD; the anthropometric and clinical variables in these groups were in risk ranges and higher compared to the individuals from the control group.

It is worth mentioning here that individuals from the MetS-CHD^+^ group had slightly higher values of all anthropometric and clinical variables except HDL as compared with subjects from the MetS-CHD^−^ group. Moreover, individuals from the case group had lower total cholesterol, LDL, HDL, and triglycerides compared with those from the control group. Our result is in support of previous reports that individuals with MetS have persistently higher risk levels of common anthropometric and clinical disease markers, which could lead to health-threatening disease conditions like CHD [5,6].

The frequency of all individual cardiometabolic risk phenotypes was very high in the combined case group. The highest frequency (76.7%) was of dyslipidemia in the disease group individuals followed by GO (56.2%), hypertension (53.4%), and CO (44.5%). The high frequency of cardiometabolic risks in Pakistani subjects also supports the previous finding that the clustering of cardiometabolic dysregulations could lead to MetS and further complications, especially CHD [4,5,6]. Genetic relationships between metabolic dysregulations and the development/severity of cardiovascular diseases have also been reported [7,8], especially the rs1333049 SNP of the 9p21 locus [29,30,42]. Several studies have also explored genetic relationships between the rs1333049 polymorphism with cardiometabolic risk components and CHD [43,44,45,46,47].

We report a significantly higher frequency of the minor C allele of the rs1333049 variant in both the MetS-CHD^+^ and MetS-CHD^−^ disease groups as compared with the healthy controls (*p* ≤ 0.0001). The association of the minor C allele was examined with MetS with or without established CHD and also with individual metabolic risk phenotypes; GO, CO, dyslipidemia, hypertension, and hyperglycemia (Table 2). The genetic association results showed that individuals from all disease phenotype groups carrying minor/risk C allele had higher disease risk ranging from OR = 2.24–4.04. Among all disease groups, a highly significant association (*p* = 0.0003) of the rs1333049 variant was found with MetS/MetS-CHD^−^ showing C allele carriers at a 4.04 times higher risk of MetS compared with non-carriers (OR = 4.04, 95% CI = 1.62–10.07). The rs1333049 SNP also showed a significant association with the CHD group (MetS-CHD^+^) but was 1.34 times lower (OR = 2.70, 95% CI = 1.16–6.28) compared with the MetS group (OR = 4.04). The contribution of the C allele towards disease risk was clearly identified from the comparison of study variables (mean ± SD) among C allele carriers versus non-carriers of the rs1333049 variant (Figure 3). The anthropometric variables (age, weight, SBP and DBP), adiposity indices (BMI, WC, WHR, VAI, BAI), and clinical test values (FBG, TG, TC, HDL, LDL) showed HDL levels were within risk ranges in C allele carriers (GC + CC) as compared with non-carriers (GG).

The genetic association of rs1333049 has been largely explored with cardiovascular diseases and complications of the heart in different world populations including Pakistan [29,30,42]. Since cardiovascular complications strongly correlate with MetS or individual cardiometabolic complications [43,44,45,46], establishing a direct association of the rs1333049 polymorphism with an already prevailing metabolic hub could uncover the exact role of the risk allele of the rs1333049genetic variant of the 9p21 locus. The present study also confirms that the minor C allele of the rs1333049 variant is a risk allele more abundant in Pakistani subjects with MetS. Our study also showed that minor C allele carriers of the rs1333049 variant had a 4 times higher risk of MetS (CHD^−^) while MetS-CHD^+^ individuals had a two-fold higher disease risk. Overall, this study reports the rs1333049 9p21 locus SNP as a strong risk predictor of MetS and individual cardiometabolic dysregulations in a Pakistani study cohort independent of CHD status.

## 5. Conclusions

The 9p21 genetic locus/rs1333049 SNP is strongly associated with, and can be a genetic predictor of, MetS and cardiometabolic risks regardless of CHD in the Pakistani population.

## Figures and Tables

**Figure 1 genes-13-02226-f001:**
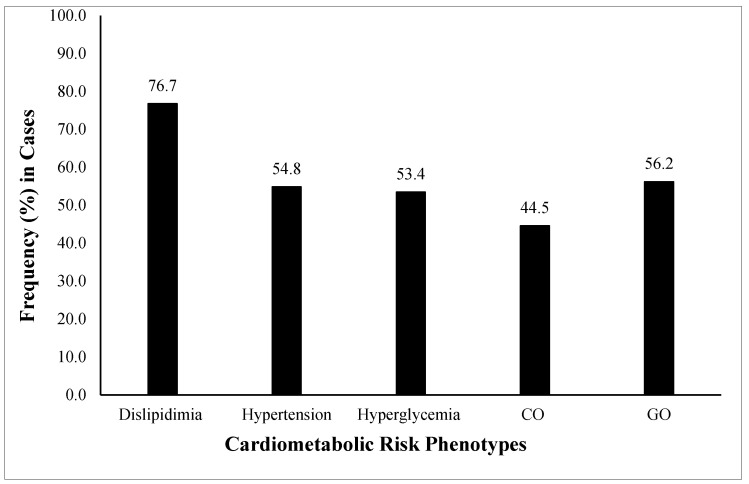
Frequency (%) of cardiometabolic risk phenotypes among the studied population. The cardiometabolic risk phenotypes presented are general obesity or GO (BMI ≥ 30 Kg/m^2^), central obesity or CO (WHR > 0.90 for males and 0.85 for females), hypertension (SBP > 139 mmHg and DBP > 89 mmHg) (34), hyperglycemia (FBG > 100 mg/dL), and dyslipidemia (LDL > 100 mg/dL, HDL < 40 mg/dL and TG > 150 mg/dL). The highest observed frequency was of dyslipidemia followed by GO, hypertension, hyperglycemia, and CO. GO, general obesity; CO, central obesity; BMI, body mass index; WHR, waist-to-hip ratio; SBP, systolic blood pressure; DBP, diastolic blood pressure; FBG, fasting blood glucose; LDL, low-density lipoprotein; HDL, high-density lipoprotein; TG, triglycerides.

**Figure 2 genes-13-02226-f002:**
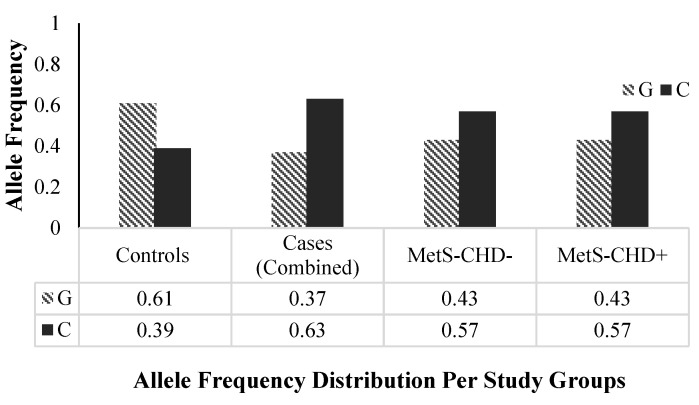
rs1333049 SNP allele frequencies distribution per study groups. The frequencies of the major “G” and minor “C” alleles of rs1333049 SNP in three study groups; healthy controls, cases (combined: MetS and CHD), Mets-CHD^−^ (MetS without CHD) and MetS-CHD^+^ (MetS with CHD) are shown. The minor “C” allele frequency is higher in cases (combined), MetS-CHD^−^ and MetS-CHD^+^ and lower in the control group. MetS, metabolic syndrome; CHD, coronary heart disease.

**Figure 3 genes-13-02226-f003:**
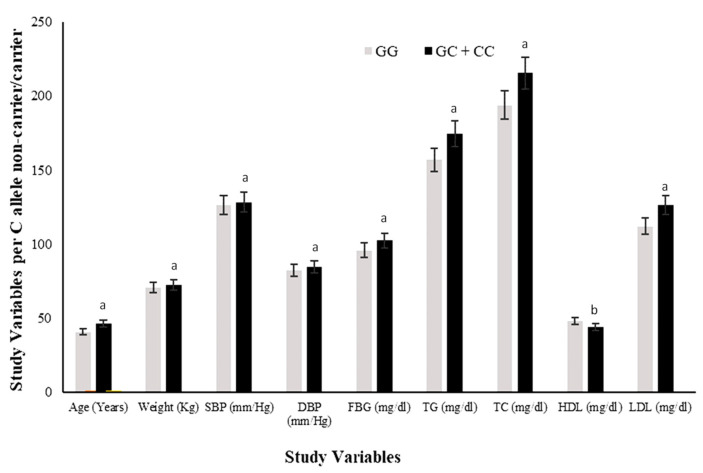
Comparison of study variables in rs1333049 SNP minor/risk (C) allele and major (G) allele carriers. Comparison of the levels of all study variables among G allele carrier (GG homozygotes) and C allele carrier (GC heterozygotes and CC homozygotes combined) study subjects. The levels of all study variables were raised in C allele carriers while HDL was lower compared with that of G allele carriers. The superscript “a” on bars for variables shows significantly higher levels in C allele carriers versus G allele carriers. Whereas superscript “b” on bars shows significantly lower levels in C allele carriers vs. G allele carriers.

**Figure 4 genes-13-02226-f004:**
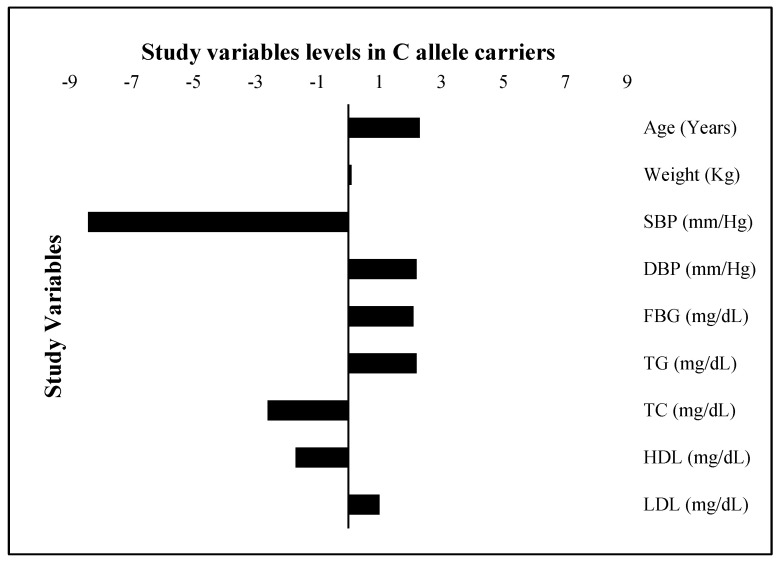
Study variable levels (SD change from population means) in C allele carriers. The levels of all study variables are presented as a standard deviation (SD) change from the population means in C allele carrier (GC heterozygotes and CC homozygotes combined) study subjects. The bars with +ive (>0) values represent higher/raised levels while those with −ive (<0) values show lower levels of individual variables in C allele carriers as compared with the G allele carriers.

**Figure 5 genes-13-02226-f005:**
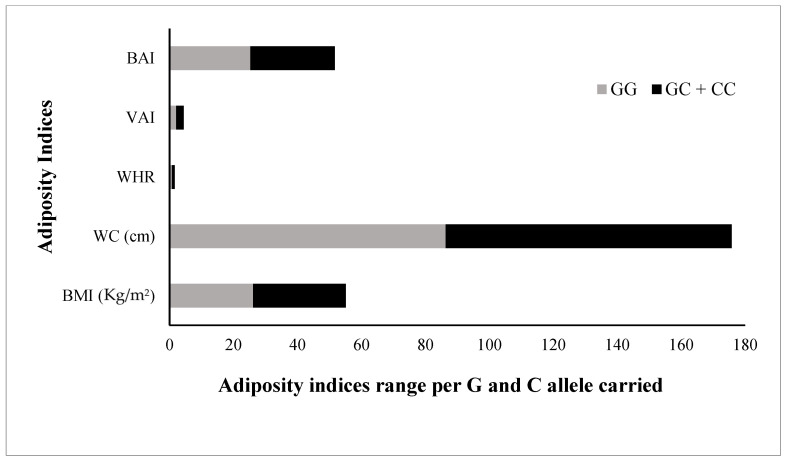
Comparison of adiposity indices in rs1333049 SNP minor/risk © allele and major (G) allele carriers. Comparison of the levels of all adiposity indices (BMI, body mass index; WC, waist circumference; WHR, waist-hip ratio; VAI, visceral adiposity index; and BAI, body adiposity index, among G allele carrier (GG homozygotes) and C allele carrier (GC heterozygotes and CC homozygotes combined) study subjects. All adiposity indices were raised in C allele carriers compared with G allele carriers.

**Figure 6 genes-13-02226-f006:**
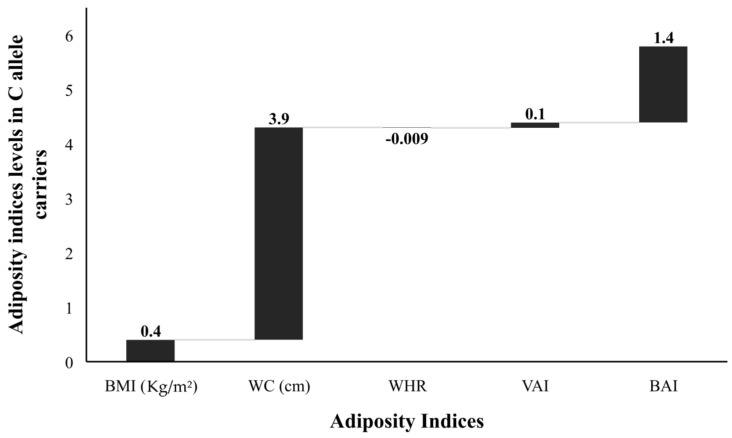
Adiposity indices levels (SD change from population means) in C allele carriers. The levels of all adiposity indices (BMI, body mass index; WC, waist circumference; WHR, waist-hip ratio; VAI, visceral adiposity index; and BAI, body adiposity index presented as standard deviation (SD) change from the population means in C allele carrier (GC heterozygotes and CC homozygotes combined) from that of G allele carrier (GG homozygotes) study subjects. The bars with +ive (>0) values represent higher/raised levels while those with −ive (<0) values show lower levels of individual variables in C allele carriers as compared with the G allele carriers.

**Table 1 genes-13-02226-t001:** Comparison of study variables among cases and controls.

Variables	Study Groups (Mean ± SD)		*p*-Value
	Control	Cases	
Age (Years)	31.1 ± 15.0	49.8 ± 16.5	7.48 × 10^−12^
Weight (Kg)	67.7 ± 9.3	73.3 ± 9.9	3.40 × 10^−4^
BMI (Kg/m^2^)	24.9 ± 2.8	29.4 ± 3.9	2.63 × 10^−13^
WC (cm)	75.6 ± 9.0	93.4 ± 15.0	3.40 × 10^−14^
HC (cm)	96.8 ± 10.7	109.0 ± 14.3	4.10 × 10^−8^
WHR	0.78 ± 0.04	0.85 ± 0.07	1.76 × 10^−10^
VAI	1.3 ± 0.5	2.7 ± 1.2	1.80 × 10^−14^
BAI	21.1 ± 4.1	28.0 ± 6.0	5.53 × 10^−13^
SBP (mm/Hg)	121.7 ± 4.2	130.0 ± 12.3	2.07 × 10^−6^
DBP (mm/Hg)	79.8 ± 4.7	85.4 ± 7.0	1.90 × 10^−7^
FBG (mg/dL)	93.7 ± 9.8	103.2 ± 15.1	2.76 × 10^−5^
TG (mg/dL)	133.6 ± 20.4	182.8 ± 47.5	4.52 × 10^−12^
TC (mg/dL)	178.6 ± 33.3	221.2 ± 38.4	1.25 × 10^−11^
HDL (mg/dL)	50.4 ± 11.6	43.1 ± 9.8	1.36 × 10^−5^
LDL (mg/dL)	108.0 ± 25.6	127.8 ± 29.4	1.98 × 10^−5^

BMI, body mass index; WC, waist circumference; HC, hip circumference; WHR, waist-to-hip ratio; VAI, visceral adiposity index; BAI, body adiposity indices; SBP, systolic blood pressure; DBP, diastolic blood pressure; FBG, fasting blood glucose; TG, triglycerides; TC, total cholesterol; HDL, high-density lipoprotein; LDL, low-density lipoprotein.

**Table 2 genes-13-02226-t002:** Genetic association of rs1333049 SNP with cardiometabolic risk phenotypes.

Study Phenotypes	Dominant/Allele Model (CC + GC vs. GG)	*p*-Value
	OR (95% CI)	
Controls	1	
Gender (M vs. F)	1.79 (1.13–3.44)	0.034
General Obesity	2.39 (1.15–5.00)	0.017
Central Obesity	1.56 (1.26–3.68)	0.031
Hypertension	2.24 (1.11–4.53)	0.022
Hyperglycemia	0.19 (0.01–4.64)	0.29
Dyslipidemia	3.25 (2.28–6.07)	0.003
MetS-CHD^−^	4.04 (1.62–10.07)	0.0003
MetS-CHD^+^	2.70 (1.16–6.28)	0.019

Genetic association analysis of the allele C of rs1333049 SNP with cardiometabolic risk phenotypes; GO (general obesity), CO (central obesity), hypertension, dyslipidemia, hyperglycemia, and disease; MetS-CHD^+^ (MetS with CHD) and MetS-CHD^−^ (MetS without CHD) using the dominant genetic model (CC + GC vs. GG). The genetic associations are presented as OR (odds ratios) and 95% CI (95% confidence intervals).

## Data Availability

All the relevant data is presented within this publication.

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
