# Peer review of "9p21 Locus Polymorphism Is A Strong Predictor of Metabolic Syndrome and Cardiometabolic Risk Phenotypes Regardless of Coronary Heart Disease"

_genes, 2022, doi:10.3390/genes13122226_

Round 1

Reviewer 1 Report

This study shows that the 9p21 genetic locus/rs1333049 SNP is strongly associated with- and can be a genetic predictor of MetS and cardiometabolic risks irrespective of cardiovascular diseases in Pakistani population.

1.How to determine the sample size of this study?

2.For the indicators in Table 1, the indicators with significant differences between the healthy control group and the case group should be marked as significant in the table, that is, P < 0.05.

3.The font of P value in the article shall be changed to upper case and italics.

Author Response

Dear Dr., Dr. Zsolt Ronai

Editor, Genes MDPI,

Thank you for giving us the opportunity to revise our manuscript (ID: genes-2043810). We are grateful for reviewers for their time, careful review, and constructive comments. We have revised our manuscript in the light of reviewers’ comments, and added the changes asked/suggested by reviewers, which can be seen with tracked changes in the revised file.

Review Comments to the Author

The point-by-point response to each reviewer is provided below.

Reviewer #1: 

This study shows that the 9p21 genetic locus/rs1333049 SNP is strongly associated with- and can be a genetic predictor of MetS and cardiometabolic risks irrespective of cardiovascular diseases in Pakistani population.

  1. How to determine the sample size of this study?

Answer: Sample size was calculated using online OpenEpi calculator (http://www.openepi.com). We have updated the text of main manuscript at page 2, line 87. 

  1. For the indicators in Table 1, the indicators with significant differences between the healthy control group and the case group should be marked as significant in the table, that is, P< 0.05.

Answer: Table 1 has been updated and a separate column for P values has been added for indicators with significant differences between the healthy control group and the case group.

3. The font of P value in the article shall be changed to upper case and italics.

Answer: The font of P value has been changed to upper case and italics throughout the text.

Reviewer 2 Report

Zafar et al present an interesting manuscript on 9p21 Locus Polymorphism as a strong predictor of metabolic 2 syndrome and cardiometabolic risk phenotypes. The manuscript is appealing, however, needs the following change:

1. Fig 2 & 3: error bar and statistics/significance missing

2. Line 205-207: significance of the following result is not clear

“The value (higher or lower) of continuous covariates in risk allele carriers showed that C allele carriers were 2.3 years older and 0.1 Kg heavier in body weight as compared to the G allele carriers”

3.  Figure 4 to 6: control data should also be presented and individual data points would give more clarity rather than mean data bars.

4.  If a similar study were done in other countries, especially South Asian ones then would be interesting to see the result comparison in the discussion.

Author Response

Dear Dr., Dr. Zsolt Ronai

Editor, Genes MDPI,

Thank you for giving us the opportunity to revise our manuscript (ID: genes-2043810). We are grateful for reviewers for their time, careful review, and constructive comments. We have revised our manuscript in the light of reviewers’ comments, and added the changes asked/suggested by reviewers, which can be seen with tracked changes in the revised file. 

Reviewer #2: 

Zafar et al present an interesting manuscript on 9p21 Locus Polymorphism as a strong predictor of metabolic 2 syndrome and cardiometabolic risk phenotypes. The manuscript is appealing, however, needs the following change:

  1. Fig 2 & 3: error bar and statistics/significance missing

Answer: Fig. 2 represents distribution of "G" and "C" allele frequencies in each study groups rather than a comparison among groups. The figure has been updated by including its data table.

The error bar and statistics/significance for figure 3 have been added and updated in the manuscript.

  1. 2. Line 205-207: significance of the following result is not clear

“The value (higher or lower) of continuous covariates in risk allele carriers showed that C allele carriers were 2.3 years older and 0.1 Kg heavier in body weight as compared to the G allele carriers”

Answer: These results show actual changes in the variables among C allele carriers (GC heterozygotes and CC homozygotes combined) verses G allele carriers (CC homozygotes) presented in figure 4 as standard deviation (SD) change from population means in C allele carrier.  For example “C allele carriers were 2.3 years older and 0.1 Kg heavier in body weight as compared to the G allele carriers”

  1. Figure 4 to 6: control data should also be presented and individual data points would give more clarity rather than mean data bars.

Answer: The data presented is actually SD difference of population means for G allele carrier (Control) to C allele carrier (Cases). We have presented the data for figures 3-4 and 5-6 as following two supplementary tables;

Table S1:              Comparison of study variables in rs1333049 SNP minor/risk (C) allele and major (G) allele carriers and SD change from population means in C allele carriers

Table S2:              Comparison of adiposity traits in rs1333049 SNP minor/risk (C) allele and major (G) allele carriers and SD change from population means in C allele carriers

  1. If a similar study were done in other countries, especially South Asian ones then would be interesting to see the result comparison in the discussion.

Answer: The current study’s focus is on the genetic association of rs1333049 SNP with cardiometabolic risks and metabolic syndrome while previous studies conducted on South Asians were specifically on association of rs1333049 SNP with cardiovascular diseases risk.

Round 2

Reviewer 2 Report

All my comments are addressed by the authors.